# Global climate-driven trade-offs between the water retention and cooling benefits of urban greening

M. O. Cuthbert [1,2 ✉], G. C. Rau [3], M. Ekström [1], D. M. O'Carroll [2] & A. J. Bates [4]

Urban greening can potentially help mitigate heat-related mortality and flooding facing the >4 billion urban population worldwide. However, the geographical variation of the relative combined hydrological and thermal performance benefits of such interventions are unknown. Here we quantify globally, using a hydrological model, how climate-driven trade-offs exist between hydrological retention and cooling potential of urban greening such as green roofs and parks. Using a Budyko framework, we show that water retention generally increases with aridity in water-limited environments, while cooling potential favors energy-limited climates. Our models suggest that common urban greening strategies cannot yield high performance simultaneously for addressing both urban heat-island and urban flooding problems in most cities globally. Irrigation, if sustainable, may enhance cooling while maintaining retention performance in more arid locations. Increased precipitation variability with climate change may reduce performance of thinner green-infrastructure more quickly compared to greened areas with thicker soils and root systems. Our results provide a conceptual framework and first-order quantitative guide for urban development, renewal and policymaking.

[1] School of Earth and Environmental Sciences, Cardiff University, Cardiff, UK. [2] School of Civil and Environmental Engineering, The University of New South Wales, Sydney, Australia. [3] Institute of Applied Geosciences, Karlsruhe Institute of Technology, Karlsruhe, Germany. [4] School of Animal, Rural & Environmental Sciences, Nottingham Trent University, Nottingham, UK. ✉email: cuthbertm2@cardiff.ac.uk

Cities are hot spots for future population increase and already home to more than 55% of the 7.6 billion global population[1]. Urban areas also have unique climates and resource demand patterns, re-shaping local water and energy budgets relative to periurban and rural environments. Well-known urban climate risks are the urban heat island (UHI) effect where the unique urban signature of sensible and latent heat fluxes interact to enhance warming[2–5]. In addition, the urban stream syndrome (USS) reflects the widely reported degraded physical, chemical, and biological conditions caused by urbanisation of watersheds[6,7] and the consequent increase in flood risk[8]. In combination, these present pressing health and water management challenges associated with substantial financial (e.g., infrastructure) and social (e.g., human wellbeing and mortality) risks, and increasingly so under climate change[7,9] due to the impact on heat and moisture fluxes.

To combat these problems, a commonly proposed strategy for urban development and renewal[10] is to increase the proportion of 'urban greening' in the form of green roofs, green walls, or vegetated urban spaces[2,11]. This can potentially reduce the UHI and USS effects and support local ecosystem services and resident wellbeing[12]. Owing to the multiplicity of potential advantages and disadvantages associated with green surfaces, multidisciplinary studies have been used to assess their overall value[13]. Although other aspects such as increased urban wildlife habitats and regulation of building temperatures are also prevalent, the mitigation of the stormwater retention component of the USS and the UHI effect are reported to be the most important associated benefits and opportunities[14].

Antecedent conditions strongly influence the ability of vegetated surfaces to delay and reduce run-off generation. The maximum ability of a vegetated surface to retain water during and after rainfall requires a period of relatively dry conditions preceding a rainfall event. This enables more potential storage to accumulate in the pores of the substrate/soil layer[13,15], driven by the loss of soil moisture back to the atmosphere. Cooling due to urban greening also occurs predominantly via increased evapotranspiration[3,4] but requires consistently moist soil conditions either from precipitation or irrigation, along with a significant potential evaporation demand. Despite the overall similarities in the driving processes for potential retention and cooling, no large-scale studies have systematically addressed the assessment of these environmental benefits in combination or their relative performance across different climatic regions. This is of critical importance since, owing to the various potential interactions between different urban greening interventions, to consider any one in isolation risks unknowingly providing a potential disservice from another.

Here we propose a conceptual, data-driven, screening-level analysis to understand and quantitatively estimate the comparative retention and cooling potential of urban greening across a global geographic extent. We apply a parsimonious hydrological model forced with climate observations and re-analysis data to estimate metrics of relative cooling and retention performance over the long term as well as their seasonal dynamics. We find that the Budyko[16] hypothesis, an empirical relationship between the long-term combined water and energy balance of catchments, is a powerful framework in which to examine and expose the trade-offs amongst the adaptive functionality of urban greening. Our results show that, globally, cities are more likely to benefit from one or other of the beneficial effects of urban greening to address either excess warming or flooding, but rarely both. Adequate local-scale information is not yet available in most locations globally to support urban development policy. Our results thus provide a much needed first-order, coarse-grained, geographic foundation that may guide land-use policy toward

multiple aims. The results can be used to broadly scope the potential of urban greening to mitigate for heat and flood-related climate risks. This is particularly valuable where local-scale information is unavailable.

## Results

**Aridity controlled retention-cooling trade-offs.** We applied a parsimonious hydrological model (Supplementary Figs. 1 and 2) worldwide forced by daily precipitation (P) and potential evapotranspiration (PET) derived from re-analysis data[17] to derive gridded global outputs (Supplementary Figs. 3 and 4). In order to make urban-specific inferences from the results, we then extracted modelled outputs for the locations of 31,500 urban areas globally from the Global Rural-Urban Mapping Project (GRUMP)[18]. To assess differences in modelled output using gridded and point forcing data, an additional suite of models was run using meteorological data from the Global Surface Summary of the Day (GSOD)[19] archive for 175 cities that contain precipitation measurements of sufficient quality and duration (see locations in Fig. 1B,E). Three urban greening types were modelled, representing extensive (depth $h = 50$ mm), intensive ($h = 150$ mm) and deep ($h = 1000$ mm) substrates or soil profiles using parameters appropriate for typical, well-drained, engineered green surfaces. Irrigated scenarios were also explored for the intensive substrate.

We used Monte Carlo experiments to resolve the parameter equifinality of the model and estimate the uncertainty in constraining our global results to a limited number of parameter sets. The model simulates drainage (D) which can be used to determine hydrological retention aggregated as (P-D)/P. Since the model also generates estimates of actual evapotranspiration (AET), which is the dominant driver for urban cooling[3,4], we also derive and evaluate a semiempirical potential cooling metric (AET/PET). This model can explain >70% of the variance in global mean UHI data[20,21] with climate for large cities (Supplementary Fig. 5). In combination, these dimensionless metrics represent a practical way of approximating the relative potential performance of retention and cooling for different urban greening interventions at any given location, or the same urban greening intervention at different locations. Our model results apply to any homogeneously vegetated surface from canopied areas as well as deeper soil profiles common in parks and recreation areas, to much more shallow and highly engineered substrates such as green roofs. The internal consistency of input data and methods enable a visualization of global patterns in the efficacy of cooling and retention solutions.

Here we make the observation that the equation for mean retention can be expressed in identical terms as the ordinate axis (AET/P) of the Budyko curve[16]. This leads logically to using the Budyko hypothesis, namely that the Earth's surface long-term water and energy balance is largely dependent upon an aridity ratio, as a framework in which to analyze the results and generalize the controls on the hydrological performance. The climate metric of relevance as the dependent variable in our analysis then becomes the abscissa of the Budyko curve (PET/P) which is also the inverse of the aridity index (P/PET). We show that the application of the Budyko framework in this way is a powerful tool for explaining the overall trade-off between cooling and retention as follows, with reference to our results for the intensive substrate (Fig. 1).

Towards the left side of sub-panels C and F in Fig. 1 where PET/P < 1, conditions are defined as 'energy limited' since there is more than enough precipitation to meet the energy demands of the atmosphere driving evapotranspiration. In such an environment, the retention capacity is lowest since only a proportion of

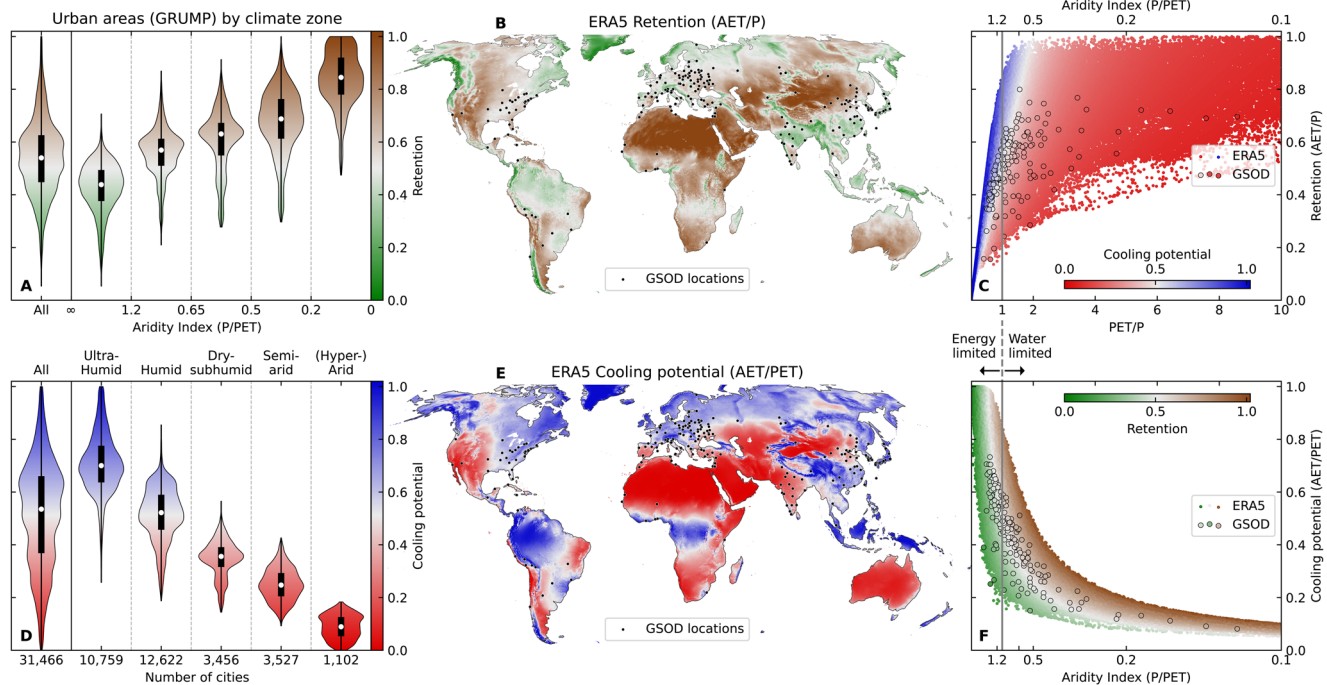

**Fig. 1 Global patterns of urban greening hydrological retention and cooling potential.** Spatial distributions and interrelationships of ERA5 re-analysis data forced models using intensive substrate ($h = 150$ mm) for: (**B, C**) Retention and (**E, F**) Cooling potential. (**A, D**) Violin plots for data extracted from GRUMP[18] urban areas only. GSOD[19] city point-data and locations are shown for reference in (**B, C**) and (**E, F**). Equivalent results for other substrates are given in Supplementary Figs. 6 and 7.

the precipitation can be evaporated, and drainage rates will be high from excess precipitation. However, cooling is principally driven by the change of sensible to latent heat during evaporation. Hence, the highest rates of cooling potential are found in the energy-limited part of the Budyko curve. This is consistent with the overall shape of global UHI observations, as should be expected, which indicate a decreasing UHI effect with increasing aridity (Supplementary Fig. 5).

In contrast as we move towards the right side of the Budyko space where PET/$P > 1$, conditions are 'water limited' since there is a limited amount of water available ($P$) for evapotranspiration (AET) despite the high potential evaporative demand from the atmosphere (PET). At this end of the Budyko curve, we have higher rates of AET as a proportion of $P$ leading to higher retention rates. But at the same time the relative amount of AET as a proportion of the overall energy budget is decreasing so the rates of relative cooling potential decrease. The general pattern is thus that, in broad terms, energy-limited climates favour cooling more than retention, whereas water-limited climates favour retention more that cooling. However, there are many important and interesting variations within this overall pattern that can be quantified from our results.

**Influence of substrate choice and irrigation on comparative retention-cooling performance.** We define a breakpoint value for the retention and cooling performance metrics of 0.5 and find that the percentage coverage of the global landmass with modelled retention performance (defined as mean AET/$P$) above the breakpoint is 44% for extensive profiles (depth $h = 50$ mm), 66% for intensive profiles ($h = 150$ mm) and 82% deep profiles ($h = 1000$ mm) (Supplementary Figs. 6–8). The respective percentage coverage for the cooling potential of (defined as mean AET/PET) above 0.5, is 43, 57, and 66% (Supplementary Figs. 6–8). Budyko curves (Eq. 10) and their equivalent cooling

potential curves describe well the upper envelopes for the simulated range of substrate/rooting depths for the GSOD cities (Supplementary Fig. 9). However, deviations below these envelopes occur increasingly for higher coefficients of variation in precipitation; the same effect is also seen within the mean across all GRUMP urban areas (Supplementary Fig. 9). This is intuitive hydrologically since, for most soils and land-cover types, larger or more intense rainfall events can more easily overcome antecedent soil moisture deficits, leading to increased drainage and hence lower retention. Since lower retention leads to lower AET, a similar, although less pronounced, the effect is also seen in the cooling potential curves. These effects are stronger the thinner the substrate/rooting depth becomes. Thus, while retention shows a strong positive trend with aridity (defined as PET/$P$) (Fig. 1A), the cooling potential shows the opposite trend (Fig. 1D), and a consistent inverse relationship between retention and cooling potential is seen across different substrate/rooting depths (Fig. 2B, Fig. 3A).

The retention-cooling relationship can be clearly visualized in the latitudinal summaries (Fig. 2B) with the steepest variations occurring across the tropics (delineated in Fig. 2), home to nearly half of the global population[22]. For intensive urban greening, over two-thirds of the landmass fails to achieve simultaneous retention and cooling potential performance above the 0.5 breakpoint. This is also the case when considering just those areas that are currently urbanized (as defined by the GRUMP city database— see Methods). However, there are some parts of the world, most notably western equatorial Africa and southwest China, in which both cooling and retention may perform well above the 0.5 breakpoint. Such locations are characterized by energy limited (i.e., where energy rather than water is the limiting factor on rates of evapotranspiration), high humidity, conditions with associated relatively low coefficients of variation in daily precipitation (Fig. 2D). Conversely, poorer performance in both metrics (i.e., under the 0.5 breakpoint) is seen in certain locations (e.g.,

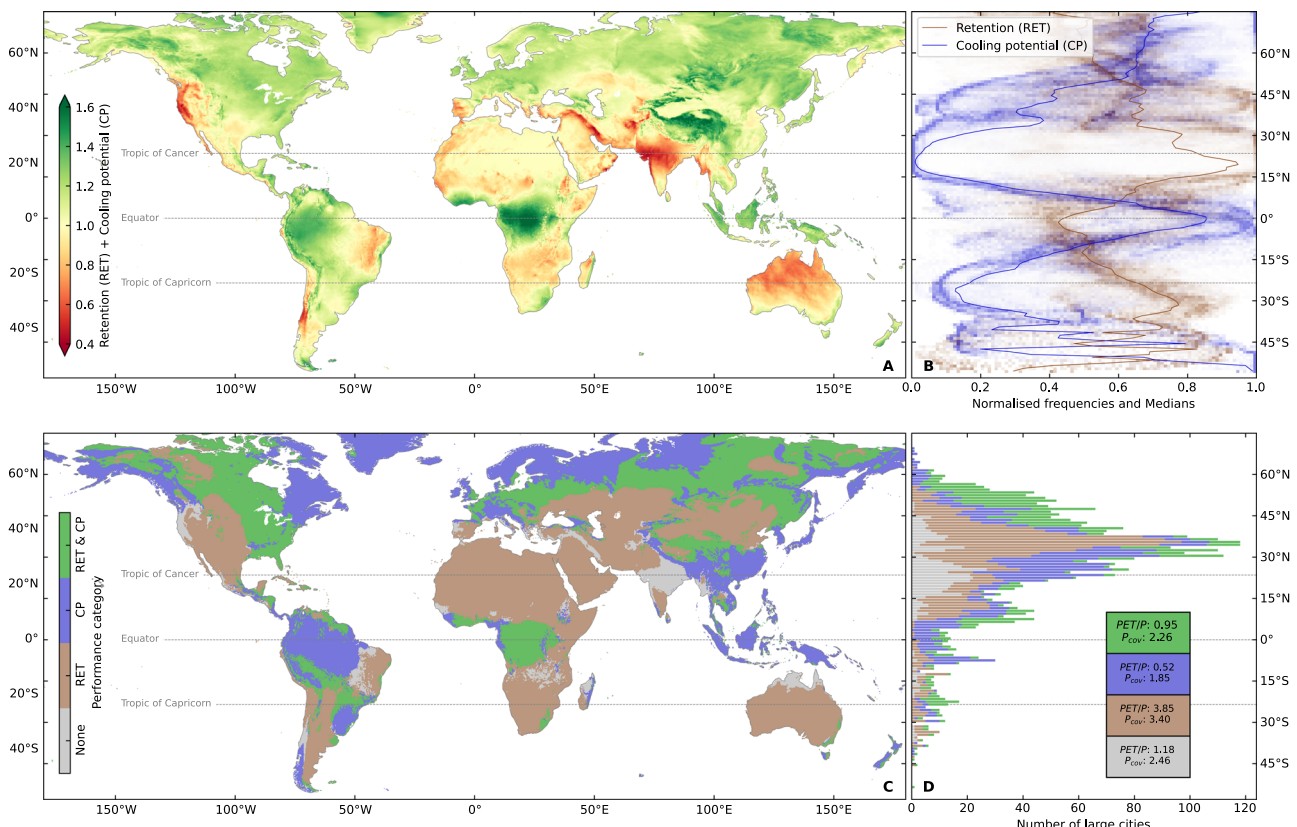

**Fig. 2 Global trade-offs between mean retention and cooling potential. (A)** Global map of the sum of mean retention and cooling potential metrics. **(B)** Latitudinal summaries of mean retention and cooling potential. Shading maps the binned frequencies normalized for 1° latitude bands. Solid lines are medians across the latitudes. **(C)** Categorized performance of mean retention only >0.5 (RET), mean cooling potential only >0.5 (CP), mean retention and mean cooling potential both >0.5 (RET & CP), or neither mean retention or mean cooling potential >0.5 (None). **(D)** Latitudinal summaries by categories defined in **(C)** for large cities (defined as having population > 100,000 in the year 2000 CE). Inset indicates the median value of PET/P and coefficient of variation in daily precipitation (Pcov) for each mapped category. All results shown are for intensive substrates ($h = 150$ mm)—equivalent results for other substrates are given in Supplementary Figs. 10 and 11.

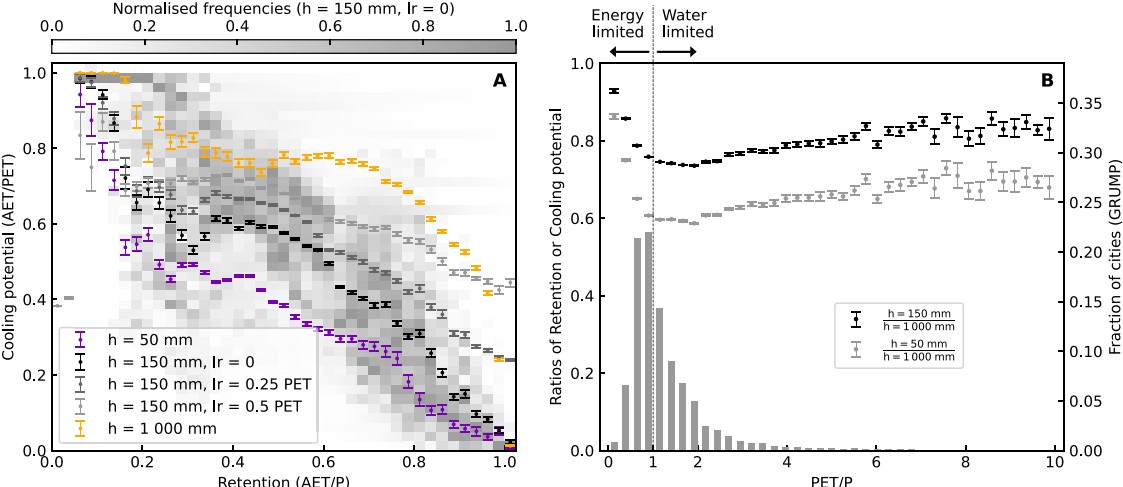

**Fig. 3 Summary of the trade-offs between mean retention and cooling for different types of urban greening. (A)** Trade-offs between retention and cooling potential for different substrate thicknesses ($h$) and irrigation ($Ir$) applications. Points are means +/− standard error of the mean (SEM), shadings represent the binned and normalized frequency values for the intensive unirrigated substrate. **(B)** Histogram of global cities binned by PET/P alongside the fraction of retention or cooling potential offered by intensive substrate ($h = 150$ mm) and extensive substrate ($h = 50$ mm) as a fraction of the deep substrate ($h = 1000$ mm). The ratio of retention and cooling potential metrics is identical as both are equivalent to the ratio of AET between the respective substrate types. Values plotted as means +/- SEM.

northern India, northern Australia, and western North America) associated with moderate (e.g., sub-humid to humid) aridity conditions. This trade-off between retention and cooling potential is more pronounced for thinner substrates but, when irrigation is included, the strength of the trade-off between retention and cooling lessens substantially (Fig. 3).

When irrigation is applied as a proportion of PET on days with no rain, negligible change is seen in mean retention values as calculated using Eq. 8. This is as expected since irrigation is always supplied at a rate less than the PET demand. Hence AET is affected, but not drainage rates, since no drainage can occur on days when PET exceeds the precipitation. However, cooling potential increases with an enhancement of AET. This is most pronounced in regions of higher aridity (Supplementary Fig. 12) where cooling potential increases of nearly 0.5 can be seen, with this effect being greater the higher the rate of applied irrigation. This suggests that irrigation may be an effective means of substantially increasing the potential cooling impacts of urban greening in areas where retention is already high, without negatively impacting the retention performance (Fig. 3A). In this way, the magnitude of the retention-cooling trade-off has the potential to be substantially reduced. In contrast, areas with low retention tend to be found in energy-limited environments where potential cooling may be substantial. In such locations, to increase retention while maintaining the high rates of cooling potential, the best option would rather be to increase the thickness, or other equivalent changes to the substrate water retention capacity, of urban greening substrates, i.e., to increase areas of open parks where deeper soils are possible.

For intensive or extensive urban greening, the minima of the retention performance or cooling potential plotted as a fraction of the deeper substrate simulations is close to the peak in the global city frequency histogram. This occurs around $PET/P = 1$, at the transition from energy-limited to water-limited hydrological conditions (Fig. 3B). This indicates that, in average terms, strategies for urban development and renewal which create green spaces with deeply rooted vegetation offer the most potential cooling or retention as might be expected. However, extensive green substrates with just 5% (i.e. 50 mm) of the thickness of the deeper substrate may already offer around 60-90% of the possible cooling or retention gains. For intensive green substrates around 75–90% of the gains may be made for soil depths of just 15% of the deep substrate. Hence a combination of such urban renewal measures may be effective in mitigating the UHI effect, or the flooding and altered hydroecological aspects of the USS, where greening large areas of urban spaces is possible. Further, our results indicate that in any given urban environment, there will be an optimum balance to be found between the performance of a particular urban greening solution, and its cost of implementation.

**Seasonality of retention and cooling potential trade-offs.** We have focused so far on mean retention since this is a commonly reported metric in the urban greening literature. However, the seasonal sensitivity in the performance of urban greening to the interactions of time-variable precipitation and PET forcings may also be important. We have therefore calculated the seasonality in retention characteristics for all GSOD city locations for the intensive substrate, preferring the point precipitation data over the gridded data owing to the increased importance of accurately reflecting the precipitation intensity variations for this purpose. We find a strong linear relationship between retention seasonality (range of monthly means) and mean retention which holds globally across climate zones (Fig. 4). This indicates that regions which are prone to low retention on average also suffer from a greater variability in retention throughout the year. In contrast,

locations with higher mean retention also have more consistent retention performance. This result is consistent with the results noted above, where locations with more variable rainfall have generally lower mean retention. However, this seasonal analysis also indicates that the greater rainfall variability is also reflected in greater seasonal retention variability. This illustrates the importance of the careful design of local flood risk management strategies in areas prone to extreme rainfall events beyond the level of detail our large-scale results can inform.

When the impact of seasonality in calculated AET is considered in regard to the cooling potential, we find that locations with high seasonal variability in cooling potential also tend to be those with higher mean cooling potential. This result stems from the interaction of timing differences between seasonal rainfall and temperature patterns in a given location. Where rainfall and temperature are strongly out of phase with each other, there is generally less variability in cooling potential than when these climate drivers are in phase with each other. Hence, these results suggest that urban greening may be more effective in locations suffering from urban heat stress where the wet season also coincides with the summer high temperatures, or where rainfall is evenly distributed throughout the year.

The spatial patterns of trade-offs we discover between retention and cooling potential, therefore also translate into the temporal domain. In general terms, high retention-low cooling potential areas will tend to also have more stable performance for each metric. Conversely, at the other end of the retention-cooling trade-off, more variable performance will generally occur in the low retention-high cooling potential areas (Fig. 4).

## Discussion

Various climate metrics are available to potentially test the degree of climatic control on urban greening performance. With respect to hydrological efficacy, previous work indicates that the Köppen-Geiger climate classification is a poor diagnostic indicator, at least for retention of green roofs[23] within the main climate classes. In contrast, our results show that aridity (i.e., $PET/P$, or its inverse the aridity index: $P/PET$) is a robust indicator of the relative performance of a given urban greening intervention since it better encapsulates the water or energy limiting nature of the prevailing climate, which ultimately controls the mean retention characteristics. From a thermal perspective, UHIs are often compared against precipitation as the dominant climate determinant. For example, it has been proposed recently that solutions for UHI focused on increasing vegetation cover or albedo are more likely to be efficient in "dry regions" but that other options may be required for "tropical cities"[4]. However, here we show that aridity ($PET/P$) may be a more useful integrative explanatory variable than precipitation alone, and hence that the Budyko hypothesis is a powerful explanatory framework in which to consider urban cooling interventions. Thus, while the cooling potential may indeed be bounded in some water-limited tropical cities, many highly populated parts of the tropics are energy limited with high cooling potential (Fig. 1, Supplementary Figs. 6 and 7). As such, cooling via increased greening may be a viable policy option unless it becomes undesirable from a comfort perspective due to subsequent elevated humidity[24].

Our results also lead to the important practical implication that irrigation in more arid locations can hypothetically improve the relative potential cooling performance while also maintaining substantial retention performance (Fig. 3, Supplementary Fig. 12). However, in many arid areas, a sustainable source of irrigated water may be a fundamental constraint to the applicability of such urban greening interventions as a cooling strategy. In general terms, groundwater is the most reliably available store of fresh water in

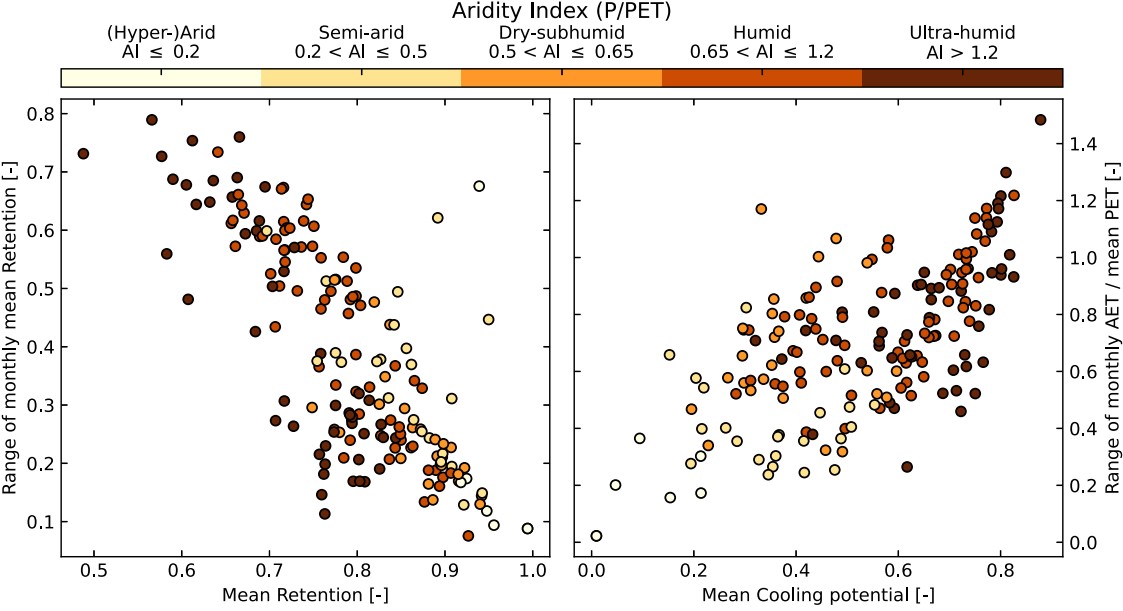

**Fig. 4 The impact of seasonality on potential urban greening performance.** Relationships between monthly retention and cooling potential and the long-term mean value, across the climatic aridity gradient, based on results from the GSOD city locations.

dryland environments, but since its replenishment may be low under the current climate[25] the water sustainability of such schemes must be considered in an integrated way to avoid unsustainable long term mining of groundwater resources[26]. Alongside urban development and renewal measures, one way of achieving this may be through the re-cycling of 'greywater', i.e. domestic wastewater[27]. It should also be noted that irrigation cannot add additional cooling gains in energy-limited environments.

While temperate and cold climates may have less need for cooling in general terms, the increased number of droughts and heatwaves is still a major concern as the climate changes[24,28]. Although there is some evidence that global warming may have complex impacts on UHI effects[29], one of the most definitive and detectable hydrological changes of global warming is the increase of precipitation intensity[30]. In this context, our results indicate that if the precipitation coefficient of variation becomes increasingly variable the modelled hydrological and thermal performance of urban greening interventions would both diminish (Supplementary Fig. 9). Further, this effect would be more pronounced for thinner, extensive soils/substrates than for thicker and more deeply vegetated surfaces. However, we also acknowledge that the exact expression of precipitation changes under climate change will also be modulated through interactions with the wider urban landscape so we consider these results to be indicative, and the need for more detailed observationally constrained local modelling is needed.

Owing to increasing population pressures and existing climate risks, urban areas play a critical role in the mitigation and adaptation to climate change; in terms of driving emissions and providing testbeds for the implementation of climate adaptation technologies and infrastructure[31]. Urban greening strategies have previously been associated with a wide range of potential environmental, economic and social ecosystem services[13,14] and while our results show that they may be effective in contributing to urban development and renewal, they are not a panacea. For example, despite potentially being able to reduce the intensity of the urban stream syndrome (USS) through their favorable effects on stream hydrology, green roofs potentially exacerbate the USS through pollution[32,33]. Hence, while the trade-offs we focus on here between retention and cooling may be important but

manageable aspects of sustainable urban development and renewal programs, other environmental trade-offs need to be considered on a case-by-case basis.

Our results show that the ability of urban greening to simultaneously mitigate local flooding or excess heat will depend strongly on the ambient local climate and substrate thickness of green practices and their overall geographic extent, urban development pattern, and renewal interventions. In general, thicker soils/substrates will provide greater benefits for both cooling and retention owing to larger soil moisture storage for evapotranspiration, although thinner substrates, such as green roofs may still provide reasonable performance. It is beyond the scope of this paper to quantify the cost-benefit analysis of different potential urban greening strategies. However, our results show clearly that there will be local optima to be sought in the balance between performance, cost and/or engineering viability in a given urban context.

While the model results are appropriate to conditions common in urban parks, recreation areas or engineered green roofs, they are not appropriate to infer to all cases of urban greening interventions (e.g., living walls or pillars, dispersed shrub, or tree planting). The gross landform and land cover of a given cityscape may add complexities not captured in our coarse-scale global analysis, e.g., due to variations in city morphology and anthropogenic heat contributions, or more complex short-term (e.g., diurnal) temporal dynamics or spatial effects of heterogeneous vegetation density[34]. However, our results are directly applicable to real-world urban greening scenarios even down to small, point, scale, as long as the application substrate is sufficiently analogous to one of those assumed by our model, and that the local climate is not significantly different to that of the climate simulated by ERA5. Since the necessary high-resolution data and modelling are not readily available for most cities around the world, our framework can provide a first-order guideline to inform generalized or large-scale strategies for urban development and renewal. We have provided our results as a lookup table (see Data Availability statement for link) for all global city areas, as a resource for such first-pass assessment as part of a tiered approach to urban development strategy until more local data or models are available with which to assess the concept at local-scales for a

particular city. Furthermore, our data can be used to explore how the performance of a green infrastructure in a current climate may perform under future climate conditions by finding a spatial analogue for a projected climate future (i.e., a space-for-time substitution).

## Methods

**Global gridded data processing, modelling, and analysis.** Global coverage of physically consistent weather variables used for modelling the hydrological and thermal performance of urban greening was obtained from the fifth generation European Medium Weather Forecast (ECMWF) atmospheric re-analysis, ERA5[17,35]. ERA5 outputs hourly forecast and analysis fields on a horizontal resolution of 31 km on 137 vertical levels (surface to 0.01 hPa)[17]. The results contain modified Copernicus Climate Change Service information 2020. Neither the European Commission nor ECMWF is responsible for any use that may be made of the Copernicus information or data it contains.

To estimate the potential evaporative water loss, we used the reference evapotranspiration (PET) as defined by Eq. 6 of the Food and Agriculture Organization (FAO) of the United Nations' Irrigation and Drainage Paper No 56 (ref. [36]). This formula, based on the Penman-Monteith method, is adapted to be representative for a hypothetical crop surface with assumed height of 0.12 m, surface resistance of 70 s m$^{-1}$, and an albedo of 0.23 as follows:

$$\text{PET} = \frac{0.408\Delta(R_n - G) + \gamma\frac{900}{T+273}u_2(e_s - e_a)}{\Delta + \gamma(1 + 0.34u_2)} \quad (1)$$

Estimates were calculated using ERA5 output (Table 1) for 2000–2017. Hourly re-analysis variables, downloaded with a 0.25° spatial resolution, were subsequently aggregated/averaged (as appropriate), to give a daily time step required for the hydrological modelling.

**Global cities data processing, modelling, and analysis.** To model the hydrological and thermal performance of urban greening using representative field datasets, urban areas with appropriate long-term Global Surface Summary of the Day[19] precipitation measurements were identified from those cross-checked for quality and applicability to large urban areas for hydrological application by Mishra et al. (ref. [37]). The datasets for 214 stations were downloaded from the National Oceanic and Atmospheric Administration (NOAA), unpacked and assembled as continuous daily time series. Missing values were flagged, and units were converted to International System of Units (SI) before further processing. Precipitation gaps smaller than 370 days were infilled using data from the Global Precipitation Climatology Centre[38] from the longitude/latitude cell within which the station is located. This resulted in a median and maximum number of infilled days of 1.57 and 1095, respectively. Data for 175 cities having at least 15 years duration to overlap with the ERA5 data (for years 2000–2015) were retained and used as input files for simulating the water retention and cooling performance of urban greening as described below.

**Urban greening model description, evaluation, and global application.** Soil moisture balance models (SMBMs) are standard tools in hydrology for simulating evapotranspiration and drainage, for example in the context of estimating crop irrigation water requirements[36], evaluating moisture deficits in soils[39,40], and estimating deep drainage (i.e. groundwater recharge)[25]. Hence, SMBMs may be confidently applied in the context of deeply rooted vegetated urban areas such as parks and recreation areas. More recently SMBMs have also been successfully evaluated for quantifying the water balance of engineered green substrates as specifically applied to urban greening measures such as green roofs across a range of climates[23,41–43]. As such, they are a well-understood and demonstrably robust tool for simulating the hydrology of a wide variety of planted green urban surfaces under diverse climatic settings with or without irrigation. However, existing modelling studies have so far not dealt with the

issue of equifinality[44], in which there may be multiple parameter sets providing equally good or acceptable model outputs. This is important to consider in a study of this kind to ensure the choice of model parameters applied at the global scale does not bias the overall results. Hence, here we first developed a standard but parsimonious SMBM. We then evaluated it using field data across a range of climates with a Monte-Carlo-based multiparameter uncertainty analysis to estimate the likely error involved in the parameter choices, before applying it globally. We recognize that urban areas are incredibly heterogeneous, with vegetation occurring in multiple assemblages, different configurations (e.g., on and under hard landscaping) and with different structural properties. However, our intention here is to provide a first order analysis at large scales and therefore this level of detail cannot be resolved in our approach here.

*Model Description.* The total available water (TAW) is defined as:

$$\text{TAW} = h(\text{FC} - \text{WP}) \quad (2)$$

where $h$ is the effective depth of rooting (m), FC is the field capacity (m$^3$/m$^3$) and WP is the wilting point (m$^3$/m$^3$). Since TAW is proportional to both $h$ and FC-WP, and thus influence the model sensitivity in a correlated way, only FC-WP was varied in the parameter sensitivity analysis outlined below while we fixed the effective rooting depth as equal to substrate depth. The readily available water (RAW) is a fixed proportion of the TAW:

$$\text{RAW} = p_c\text{TAW} \quad (3)$$

where $p_c$ is a constant (dimensionless). The model requires forcing time series of reference crop evapotranspiration (PET) and precipitation ($P$), and optionally also irrigation (Ir). Overland flow has been assumed to be zero with all rainfall becoming evaporation or infiltration as appropriate to well-drained engineered surfaces. The model follows the FAO56 'single crop-coefficient' approach[36] whereby the effect of both crop transpiration and soil evaporation are integrated into a combined crop coefficient ($k_c$) to estimate potential evapotranspiration (PET) demand from PET as follows:

$$\text{PET}_c = k_c\text{PET} \quad (4)$$

Vegetative stress and limits to soil evaporation under dry soil conditions are controlled using a stress coefficient ($K_s$, dimensionless) as follows:

$$K_s(t) = \frac{\text{TAW} - \text{SMD}(t)}{\text{TAW} - \text{RAW}} \quad (5)$$

where $t$ is time, SMD is the soil moisture deficit (m). If $P_t + \text{Ir}_t < \text{PET}_{c,t}$, all the rainfall and irrigation becomes actual evapotranspiration (AET$_t$) plus a further amount taken from the soil store equal to the remaining evaporative demand modified by the stress co-efficient as follows:

$$\text{AET}_t = K_{s,t}\text{PET}_{c,t} \quad (6)$$

If $P_t + \text{Ir}_t > \text{PET}_{c,t}$, then the excess precipitation reduces the SMD but if the SMD reaches zero then any further moisture excess becomes drainage ($D_t$) (this is equivalent to what is termed runoff in some green roof modelling studies). The overall mass balance equation that controls the state variable SMD is as follows:

$$\text{SMD}_{t+1} = \text{SMD}_t - (P_t + \text{Ir}_t) + \text{AET}_t + D_t \quad (7)$$

*Model testing and uncertainty analysis.* As discussed above, for the global application of the SMBM it is important to quantify the impact of equifinality on the overall uncertainty of the SMBMs for any particular parameter combination. To do this we took advantage of data from a recent unique field study in which climate and drainage data were collected for identical experimental green roofs in three contrasting Canadian climates[15]. The sites have long-term aridity index ($P$/PET) values of 0.64 (Calgary, Alberta), 1.35 (London, Ontario), and 2.34 (Halifax, Nova Scotia), representing the aridity range of more than 70% of global cities (using climate data from ref. [45] and the GRUMP global cities database[18]). On-site climate observations were used to calculate a daily time series of PET using the FAO Penman-Monteith equations (see ERA5 section above) and precipitation to drive the models, and the modelled drainage was compared to observed values.

---

**Table 1 PET input variables as estimated from ERA5 variables.**

| Input variable | FAO56 definition | Estimation |
|---|---|---|
| $R_n$ | Net radiation at the crop surface [MJ m$^{-2}$ day$^{-1}$] | Derived from ERA5 surface net solar/thermal radiation |
| $G$ | Soil heat flux density [MJ m$^{-2}$ day$^{-1}$] | Following Eq 42 in ref. [36] $G_{day} \approx$ zero |
| $T$ | Mean daily air temperature at 2 m height [°C] | ERA5 2 m temperature |
| $u_2$ | Wind speed at 2 m height [m s$^{-1}$] | Derived from ERA5 10 m $u$ and $v$ components and downscaled using Eq 47 in ref. [36] |
| $e_s$ | Saturation vapor pressure [kPa] | Estimated from ERA5 2 m temperature using Eq. 11 in ref. [36] |
| $e_a$ | Actual vapor pressure [kPa] | Estimated from ERA5 2 m dewpoint temperature using Eq 14 in ref. [36] |
| $\gamma$ | Psychrometric constant [kPa °C$^{-1}$] | Estimated using Eqs. 7 and 8 from ref. [36] using ERA5 geopotential to derive elevation above sea level |
| $\Delta$ | Slope vapor pressure curve [kPa °C$^{-1}$] | Estimated from ERA5 2 m temperature using Eq 13 from ref. [36] |

To assess the combined parameter and data input uncertainty, 10 000 Monte Carlo Experiments (MCEs) were run using the same parameters for all roofs for each simulation. Forcing $P$ and PET were run with added random noise at the 10% level to account for instrument uncertainties[46,47], and all parameters were randomly sampled from a priori parameter ranges as follows: 0 to 0.3 for FC-WP, 0.5 to 1.5 for $k_c$ and 0.3 to 0.7 for $p_c$. The thickness parameter $h$ was set to the known depth of the roof substrate (0.15 m).

Any individual model was accepted as behavioral if its Nash-Sutcliffe Efficiency (NSE) was > 0.7 with the additional criterion that the mean NSE across all sites was > 0.8. A linear regression model was also fitted post hoc to give an additional evaluation metric to compare the magnitudes of observed and simulated drainage events via the $R^2$ value. Time series and cumulative time series are shown in Supplementary Fig. 1. Runs with acceptable NSE values also all have $R^2$ values above 0.84 (Supplementary Fig. 2). This is indicative of excellent model performance both in terms of timing of drainage and cumulative drainage, and therefore also of mean retention and AET.

Finally, to assess the overall uncertainty of a particular choice of parameters for the subsequent global models, the behavioral parameter sets from the MCEs were simulated for all the GSOD city-data locations, forced using GSOD precipitation and ERA PET. The results indicate that the relative standard error in retention is less than approximately 1% across all behavioral parameter sets for PET/$P$ values from 0 to 8 i.e., ranging from hyperhumid to arid climates. This indicates that the application of our model to the range of global climates is very insensitive to a choice of particular parameters from the behavioral set. As a result, we decided to apply our best a priori estimate of parameters for a typical green roof for the global simulations as follows: $k_c = 1.0$, FC-WP = 0.12 (equivalent to a sand-loam substrate), and $p_c = 0.5$. This set of behavioral parameters works very well for the green roof evaluation ($R^2 > 0.85$) but also has the added advantage of being an appropriate choice, in combination with a larger value of substrate depth ($h$), for the simulation of grassed surfaces which tend to dominate in urban parks[36].

*Model evaluation of global applicability.* To further evaluate the realism and wider applicability of our model, we conducted a detailed review of the literature. We specifically targeted studies for which sufficient information was available to parametrise our model directly based on information stated in the paper (i.e. with no calibration), as well as retention measurements for multiple events from urban greening interventions. Unsurprisingly, such experimental discharge observations are only widely available specifically for green roof interventions, and we found 13 papers representing a total of 26 different roofs across a diverse range of climates[15,48–59]. At each of these locations, we ran the model using gridded ERA5 forcing input data for the specific date range of the individual studies and then compared the mean retention for the same time-period from each study with our model results. We found a very strong correlation ($R^2 = 0.78$) between the observed and modelled retention (Supplementary Fig. 2) within the typical range of scatter of observational uncertainties expected in such studies. This is a significant additional demonstration that our method can provide robust estimates, even at relatively small scales, of the hydrological functioning of a wide variety of green surfaces in urban areas, despite the coarseness of the forcing data and the inherent assumptions made in the modelling.

*Global model application.* To clearly visualize the relationship between the efficacy of the greening interventions and the ambient regional climate, our approach was applied to a global domain. In using physically consistent input data in our framework, we remove variability in performance (as per our metrics) due to changes in instrumentation or post-processing of datasets. For these calculations, three substrate thickness scenarios were assigned of 150 mm ('intensive' substrate), 50 mm ('extensive' substrate), and 1000 mm ('deep' substrate emulating, for example, a ground-level park or recreational area).

These scenarios were run globally using ERA5 PET and $P$ values on a 0.25° grid for the period 2000-2017 on a daily timestep. Whilst this approach allows a globally consistent estimate of PET, we acknowledge that underlying assumptions in the formulation of PET are partly violated; i.e., the assumption that input weather data are representative of surface weather conditions above a well-watered crop surface, which ideally extend at least 100 m in each direction of the weather station[60]. In using re-analysis, we also violate the assumed scale (point to areal estimate) and surface homogeneity. Because spatial disaggregation typically implies an amplification of uncertainties[61], we can assume that in using ERA5 over point measurements, we are reducing the representation of uncertainty in the estimates as we are effectively smoothing spatial variability in using coarser resolution data.

In addition to the simulations using only natural precipitation, we have run two global models which assume the addition of irrigation, to test its impact on the retention and cooling potential performance. For these runs, we used the intensive substrate model ($h = 150$ mm) as a baseline and applied additional water only on days of no rainfall, as a rate equal to either 25% or 50% of the PET for that day.

Modelled outputs were extracted from the global ERA5 runs to form a subset at the locations of 31,500 urban areas worldwide[18]. In order to test the sensitivity of the results to the use of gridded (ERA5) versus point (weather station observations) precipitation data[62], we also ran identical models for the time period 2000–2015 using both gridded and site observation precipitation data for the GSOD locations (see above). Implicit in our modelling is that local vegetation types would be chosen

to account for the local climate conditions, but for purposes of global comparisons, we assume a single $k_c$ everywhere.

## Metrics and empirical models of hydrological and thermal performance

*Hydrological retention.* As is conventional in the urban greening literature[15,48–59], hydrological retention (RET) was calculated as the difference in the incident precipitation and subsequent drainage over a given period ($\bar{D}$), normalized by the precipitation ($\bar{P}$), over the same period as follows:

$$\text{RET} = \frac{\bar{P} - \bar{D}}{\bar{P}} \tag{8}$$

Over the long timescales simulated this can also be expressed in terms of the mean AET ($\overline{\text{AET}}$), as:

$$\text{RET} = \frac{\overline{\text{AET}}}{\bar{P}} \tag{9}$$

The retention is expressed as a proportion of the incident precipitation, so we are not making any assumptions about the percentage spatial coverage in a particular location. Rather, the only assumption implicit in using this metric is that the total amount of retention that is possible will scale linearly with changes in the proportion of a certain type of urban cover. Hence this dimensionless retention metric can be used to represent the relative retention performance of (1) different urban greening interventions at any given location, or (2) the same urban greening intervention at different locations.

*Cooling potential.* It has recently been demonstrated that the prevailing climate is the main control on the magnitude of the urban heat island effects globally[3,4] primarily due to the dominant effect of changing landcover on the rates of evaporative cooling. Hence, in nonirrigated situations, we would expect that the observed urban heat island (UHI) effect on surface temperatures would be approximately mitigated by re-establishing the AET of native vegetation. This finding led us to seek a parsimonious heuristic metric of relative cooling potential (CP) which could be used for comparing the relative performance of a particular urban greening intervention between different locations globally. Initial exploration demonstrated that a simple scaling of AET (for which we used global AET data from GLEAM[63]), leads to too linear a relationship with climate as compared to existing UHI data. However, we found that also normalizing by the potential evapotranspiration rate leads to a very strong correspondence with observations (Supplementary Fig. 5). The Budyko framework can conveniently be used to describe this scaling as follows.

The commonly employed Turc (1954) formulation[64] which requires one parameter ($\gamma$) to control the Budyko curve as follows:

$$\frac{\overline{\text{AET}}}{\bar{P}} = \varnothing (1 + \varnothing^{\gamma})^{-\frac{1}{\gamma}} \tag{10}$$

where $\varnothing = \frac{\overline{\text{PET}}}{\bar{P}}$.

Considering the difference between rural and urban Budyko curves (parameters $\gamma_r$ and $\gamma_u$, respectively) for any climate, adding an empirical scaling factor ($f$, (°C)$^{-1}$)), and re-arranging, we can write:

$$\triangle T_s = \frac{(1 + \varnothing^{\gamma_r})^{-\frac{1}{\gamma_r}} - (1 + \varnothing^{\gamma_u})^{-\frac{1}{\gamma_u}}}{f} \equiv \frac{\triangle \left( \frac{\overline{\text{AET}}}{\bar{P}} \right)}{\left( \frac{\overline{\text{PET}}}{\bar{P}} \right) f} \equiv \frac{\triangle \overline{\text{AET}}}{\overline{\text{PET}}.f} \tag{11}$$

Where $\triangle T_s$ is the potential urban heat island effect (°C), and thus the maximum potential cooling afforded by complete urban greening as an UHI mitigation measure. This grows as the difference between the Budyko curves, and thus the difference in native and urban evapotranspiration ($\triangle \overline{\text{AET}}$), grows. The divisor empirically accounts for other controls on the UHI (e.g., predominantly changes in convective efficiency[4]), as well as variation of temperature sensitivities to changes in energy forcing at the land surface.

We demonstrate the effective performance of this empirical model by assuming a rural global Budyko value in the range 1.4–2.6 (ref. [65]) and factor this by a mean estimate of the global green urban proportion of 15% (ref. [4]) to yield a range of urban Budyko values. We then ran 1000 Monte Carlo realizations (tested to be more than sufficient for a convergent result) for uniform random combinations across the range of these urban and rural Budyko curves while fitting models using a single global value of the unconstrained empirical parameter $f$. We find this heuristic model can explain 68–73% of the variance in the global mean UHI data[21] (satellite-derived urban-rural temperature differences over a 15 year period) when plotted against aridity (Supplementary Fig. 5A). This is further evaluated by showing that the model explains >70% of the variance in the global UHI data when plotted against precipitation (Supplementary Fig. 5B). We note this gives similar performance to a more complex coarse-grained UHI model[4] when compared to the equivalent, but more limited, global UHI dataset that was used in that study[20] (also plotted for comparison purposes in Supplementary Fig. 5).

To use this result for the purpose of approximating the relative cooling effect as a comparator from one location to another, or between different substrate choices at the same location, we can discard the constant $f$ and define a dimensionless

cooling potential as equal to:

$$CP = \frac{\overline{AET}}{\overline{PET}} \qquad (12)$$

Given the demonstrated insensitivity of changes in $\Delta T_s$ to changes in urban green cover ($gc,u$) for any given mean precipitation[4] this metric can be considered applicable for any given change in urban green cover i.e., the reduction in the UHI can be expected to occur linearly with increasing urban green cover in a given climate.

The cooling potential metric we propose is therefore a practical approximation of the relative potential thermal performance of (a) different urban greening interventions at any given location, or (b) the same urban greening intervention at different locations. The dimensionless nature of cooling potential also makes it readily applicable for making comparisons to the dimensionless retention metric enabling us to straightforwardly combine and compare the metrics to consider the trade-offs globally using the comparative performance of each (e.g. Fig. 2).

**Comparison of re-analysis and point data**. Using gridded ERA5 precipitation data yields a bias of 3% ($R^2 = 0.87$) in cooling potential and 10% ($R^2 = 0.62$) in retention versus simulations using point GSOD city precipitation data. This is expected due to a fundamental scale discrepancy between the modelled areal estimate and point measurements[62], and influences of any structural or parametric inadequacies in the model used to generate the re-analysis. Different regional assessments of ERA5 precipitation suggest performance varies geographically and depending on the metric of interest. For example, ERA5 provides realistic estimates of water and energy budgets for the Canadian prairies[66], and is able to reproduce spatial precipitation distribution and light to medium quantities for Austria, but it systematically overestimates on monthly time scales[67]. For Bangladesh, compared to other gridded rainfall products, ERA5 shows good performance for rainfall detection metrics with low false alarm ratio metrics for the higher quantiles[68]. However, when station networks are sparse, re-analysis datasets can provide meaningful and spatiotemporally complete input datasets to hydrological modelling[69].

The strong correlations for outputs based on ERA5 and point data gives us the confidence to draw conclusions about the urban greening metrics (retention and cooling potential) and their climate relationships. Furthermore, the GSOD cities represent a subset that is consistent within the global range in the Budyko parameter space (Fig. 1C) including for potential cooling (Fig. 1F). Hence, for this study, we consider that ERA5 provides a robust and physically consistent dataset to derive global patterns on retention and cooling potential.

## Data availability

The model output data from this study are available online from https://doi.org/10.6084/m9.figshare.17049806 or on request from the first author. The raw precipitation data and climatic data used in the calculation of potential evapotranspiration which were used to force the models are available from the Copernicus Climate Change Service (C3S) Climate Data Store and described by Hersbach, H. et al., (2020)[17].

## Code availability

The hydrological model code used to generate the outputs from the study is available online at https://doi.org/10.6084/m9.figshare.17049806 or on request from the first author.

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

## Acknowledgements
We gratefully acknowledge coding support from Ian Thomas at Cardiff University. MOC gratefully acknowledges funding for an Independent Research Fellowship from the UK Natural Environment Research Council (NE/P017819/1). GCR gratefully acknowledges funding from the European Union's Horizon 2020 research and innovation program under the Marie Skłodowska-Curie grant agreement No 835852.

## Author contributions
Conceptualization: Conceived by MOC and refined by all authors. Methodology: MOC, ME, and GCR. Analysis: MOC, GCR, ME, DO'C, and AJB. Visualization: GCR, MOC. Writing: MOC with input from GCR, ME, DO'C, and AJB.

## Competing interests
The authors declare no competing interests.
