## [Peer Review File · Nature Communications]

Global climate-driven trade-offs between the water retention and cooling benefits of urban greeningREVIEWER COMMENTS

Reviewer #1 (Remarks to the Author):

Please see attached, annotated manuscript.

The authors present a coupled hydrologic - energy flux modeling method of identifying areas that may have favourable conditions for hydrologic retention (regulation of flooding events), near-surface cooling, or both types of services. Figure 2 indicates how these services may be more or less exclusive for different gross geographic expanses.

There is a possible story here, but it is buried in complicated, disorganized prose and what appears to be an incomplete rendering of the circumstances of the study. For example, the utilization of a published hypothesis (i.e., Budyko), could provide the fundamental framing of this analysis and its practical application or testing of this hypothesis. In any case, the conditions of geographically-extensive green space (as what appears to be green roof infrastructure, and then park areas with 1m topsoil depth) is not well-explained. The suggestion (as I understand it) of using limited groundwater resources to supplement soil moisture and bolster PET, AET is at best problematic in terms of sustainable water resources management.

Please refer to the annotated manuscript for extensive comments and suggestions on how to improve this manuscript.

Reviewer #2 (Remarks to the Author):

The study investigated the Global climate-driven trade-offs between the water retention and cooling benefits of urban greening. Generally, the topic is interesting and worth to explore, and the method is clear described. However, the biggest challenge is how can you make sure the accuracy of the modelling result, specifically, the global applicaiton of such results is challengable. detail comments see blow:

1. Introduction part needs to more precisely since the current version is not highlighted and focused, such as you described "Despite the overall similarities in the driving processes for potential retention and cooling, no large-scale studies have systematically addressed the assessment of these combined environmental benefits or their relative performance across different climatic regions." Actually, it already has studies investigate such questions, like "Magnitude of urban heat islands largely explained by climate and population".
2. Result Part described too many methods and it is necessary to focus on the results themselves.
3. I would recommend you re-organize the structure of the discuss part with subtitle.
4. line 240-241 How did you get this comments? or you get from reference?
5. line 249-250. I did not see clear evidence regarding this conclusion. you should explain more.
6. More explanation is needed to clarify the potential global application.

Reviewer #3 (Remarks to the Author):

This paper presents an assessment of the potential performance of 'urban greening' with respect to evaporative cooling and water retention (i.e. drainage) characteristics in different climate regimes. Much of what is reported is well known, but the authors make some useful observations about the trade-offs between these two functions in different settings which may be of interest to

the journal readership. The topic is certainly an important one and this kind of information would have value for onward use in practice, if sufficiently caveated, qualified and verified.

The general premise is logical and the results are plausible in the context of the assumptions outlined in the methods section. However, there are a very large number of assumptions made. Some of these are identified as violating the assumptions of the methods employed (e.g. line 405) which does raise questions about the robustness of the analysis overall. Furthermore there are questions about how far the methods employed adequately represent (a) urban processes, characteristics and conditions, (b) hazardous events (as the context for the work) and (c) processes associated with different forms of greening interventions.

For (a) above, the authors do pick up on a number of limitations, including the coarseness of the data inputs. For instance, data which are >31km resolution provide very crude measures of UHI, and the specific definition of and source of information on UHI needs to be clarified to understand the claims and replicate the method. Urban areas are incredibly heterogeneous, with vegetation in multiple assemblages, different configurations, e.g. on and under hard landscaping, and with different structural properties. This is not really acknowledged, though it is appreciated that it would not be possible to deal with all of these different aspects in a first order analysis of this sort. For (b), the authors have not elaborated on important intra-annual contexts for UHI, how the form of precipitation can influence runoff (and related pluvial flooding events). For (c) there is little or no consideration of the 'green elements' themselves. For instance, towards the end of the article it becomes obvious that the analysis and framing is based on the very specific context of green roofs (reference to 'substrate depth', crop height assumptions (0.12m), and the assumptions of functions being only related to evapotranspiration). This is also clear in the literature used. The extent to which the findings are therefore transferable to 'urban greening' in a general sense must therefore be much more carefully considered. For instance, the functional role of urban tree canopy would be expected to differ from the assumptions made here, including the important role of reducing the amount of solar radiation reaching the ground surface (see for instance <https://doi.org/10.1016/j.buildenv.2019.106606>). In my view it is not appropriate to infer to all cases of urban greening interventions. Although the authors make some qualifications in places, e.g. references to 'ground level parks' (line 401) in the discussion and 'engineered green surfaces' (line 91), these terms are ambiguous. The conclusions need to be tempered accordingly and in my view the paper reframed in the context of the interventions actually being considered. This would help with accessibility, since the terminology used seems strongly related to that used within the green roof literature.

As suggested above, at present the claims made in the manuscript are not fully supported by the analysis and require more qualification and explanation to be fully persuasive. This includes making inferences about flooding. Other aspects of the manuscript are also overstated such as the claim in the abstract that the "geographical variation of the relative hydrological and thermal performance benefits of such interventions are unknown". There are in fact multiple examples of such studies carried out in different climate zones including in the US, Brazil and Europe, some illustrations include DOI: 10.1175/JHM-D-15-0112.1, 10.1016/j.apenergy.2018.03.190. Still, the claim is more suitably qualified in the manuscript itself. Further comparison of results with other studies would be really valuable. Is it possible to carry out some verification of results using examples in the literature? This would make a very good addition to the paper. Scale may well be an issue here, but that also raises questions about the scale of analysis compared to the scale of interventions. The authors touch on some site-level data for Canada.

Finally, it would be very interesting to if the authors could elaborate further on the conceptual framework and 'targeted quantitative guide' mentioned at lines 31-32. These would be great additions, but it wasn't quite clear where either were delivered in the current version.

RESPONSE TO REVIEWERS' COMMENTS for:

**Global climate-driven trade-offs between the water retention
and cooling benefits of urban greening**

by: M. O. Cuthbert, G. C. Rau, M. Ekström, D. M O'Carroll, A. J. Bates.

Author responses in blue, line numbers refer to clean revised Manuscript.

Reviewer #1 (Remarks to the Author):

Please see attached, annotated manuscript.

The authors present a coupled hydrologic - energy flux modeling method of identifying areas that may have favourable conditions for hydrologic retention (regulation of flooding events), near-surface cooling, or both types of services. Figure 2 indicates how these services may be more or less exclusive for different gross geographic expanses.

Many thanks to the Reviewer for their extensive comments and many constructive suggestions.

There is a possible story here, but it is buried in complicated, disorganized prose....

We are sorry the Reviewer found our prose to be disorganized – we have now made modifications to the text to make key points clearer in line with their detailed suggestions.

...and what appears to be an incomplete rendering of the circumstances of the study. For example, the utilization of a published hypothesis (i.e., Budyko), could provide the fundamental framing of this analysis and its practical application or testing of this hypothesis.

We appreciate the suggestion to bring the Budyko framework more to the fore in the paper and have now done so, including in the abstract.

In any case, the conditions of geographically-extensive green space (as what appears to be green roof infrastructure, and then park areas with 1m topsoil depth) is not well-explained.

We have now added additional clarification as to the types of green infrastructure we are focussing on in this study in several places in the revised paper (e.g. L20, L322-324)

The suggestion (as I understand it) of using limited groundwater resources to supplement soil moisture and bolster PET, AET is at best problematic in terms of sustainable water resources management.

We agree that irrigating for this purpose is likely problematic (but is currently being done in many locations anyway) which is why we highlighted the need for this to be done with proper regard to sustainable water management. Nevertheless, we have strengthened this point in the paper to make it clear this may be a fundamental constraint as the Reviewer suggests – see Lines 292.

Please refer to the annotated manuscript for extensive comments and suggestions on how to improve this manuscript.

We have responded to the detailed review comments embedded within the ‘tracked changes’ version of the paper.

Reviewer #2 (Remarks to the Author):

The study investigated the Global climate-driven trade-offs between the water retention and cooling benefits of urban greening. Generally, the topic is interesting and worth to explore, and the method is clear described.

Many thanks to the Reviewer for their comments and constructive suggestions.

However, the biggest challenge is how can you make sure the accuracy of the modelling result, specifically, the global applicaiton of such results is challengable. detail comments see blow:

1. Introduction part needs to more precisely since the current version is not highlighted and focused, such as you described "Despite the overall similarities in the driving processes for potential retention and cooling, no large-scale studies have systematically addressed the assessment of these combined environmental benefits or their relative performance across different climatic regions." Actually, it already has studies investigate such questions, like "Magnitude of urban heat islands largely explained by climate and population".

Apologies if our text was not clear enough on this point – we are aware of research on cooling as well as retention of urban greening but our point here is that no studies have addressed their combined performance and the trade-offs between them in general terms. We have changed the text to make this clearer in Lines 61-66 to:

"Despite the overall similarities in the driving processes for potential retention and cooling, no large-scale studies have systematically addressed the assessment of these environmental benefits in combination or their relative performance across different climatic regions. This is of critical importance since, owing to the various potential interactions between different urban greening interventions, to consider one ecosystem service in isolation risks unknowingly providing a potential disservice from another."

2. Result Part described too many methods and it is necessary to focus on the results themselves.

Thanks for the suggestion. In short format papers where the Methods section comes later in the paper it is sometimes helpful to have a short Methods summary at the start of the paper sufficient for the reader to get an overview of what was done, which is why we included this here. We are happy to delete or shorten, but for now have left this in in case the Editor wants to give a steer on whether they prefer this or not.

3. I would recommend you re-organize the structure of the discuss part with subtitle.

Thank you for this suggestion. Unfortunately it appears that this is not allowed under the journal’s formatting policy <https://www.nature.com/documents/ncomms-formatting-instructions.pdf>.

4. line 240-241 How did you get this comments? or you get from reference?

Yes, this was derived from the lead authors previous work (REF 26).

5. line 249-250. I did not see clear evidence regarding this conclusion. you should explain more.

This statement is partly a reference to previous work indicating the intensification of precipitation due to climate change which is now well established in the literature (REF 31), and partly a conclusion from our work specifically in terms of the impacts of such intensification on urban greening cooling and retention. We have extended the discussion of this point in Lines 303-306.

6. More explanation is needed to clarify the potential global application.

We agree and have now added more description of this in Lines 327-226.

Reviewer #3 (Remarks to the Author):

This paper presents an assessment of the potential performance of ‘urban greening’ with respect to evaporative cooling and water retention (i.e. drainage) characteristics in different climate regimes. Much of what is reported is well known, but the authors make some useful observations about the trade-offs between these two functions in different settings which may be of interest to the journal readership. The topic is certainly an important one and this kind of information would have value for onward use in practice, if sufficiently caveated, qualified and verified.

Many thanks to the Reviewer for their comments and many constructive suggestions.

The general premise is logical and the results are plausible in the context of the assumptions outlined in the methods section. However, there are a very large number of assumptions made. Some of these are identified as violating the assumptions of the methods employed (e.g. line 405) which does raise questions about the robustness of the analysis overall.

We are glad the Reviewer accepts the overall premise and logic and agree that we have had to make a number of key assumptions including those underlying the PET calculations specifically referred to. Our intent was to conduct a global study, this therefore implies decisions around what uncertainties to accept in terms of data sources. Since we required several weather variables, we could have opted to source regional or variable specific datasets and combine them with the inherent uncertainties in sampling, gridding, and in instrumentation. However, because the priority for the study is a global comparison, we prioritised the physical consistency of using re-analysis data, but accept that it is representative of a greater spatial area, hence underestimating uncertainty on local level. Further, as we use daily data, some of the point-to-area average discrepancies will be smoothed out on this time-step. To demonstrate that we are justified in using ERA5, we have provided checks by comparing the ERA5 derived cooling potential and retention indices with point GSOD city data in section ‘Comparison of re-analysis and point data’. In addition, we note that PET is not a variable which can be directly observed or, as such, directly evaluated. In the hydrological community a wide variety of PET ‘products’ are used with the evaluation step occurring once AET has been estimated, as we have done here. As described further below we have now added additional evaluation of the model’s performance using the gridded forcing data. These results demonstrate our transparent assessment of

the potential impact of this assumption and how we have deemed the method robust for further analysis.

Furthermore there are questions about how far the methods employed adequately represent (a) urban processes, characteristics and conditions, (b) hazardous events (as the context for the work) and (c) processes associated with different forms of greening interventions.

For (a) above, the authors do pick up on a number of limitations, including the coarseness of the data inputs. For instance, data which are >31km resolution provide very crude measures of UHI, and the specific definition of and source of information on UHI needs to be clarified to understand the claims and replicate the method. Urban areas are incredibly heterogeneous, with vegetation in multiple assemblages, different configurations, e.g. on and under hard landscaping, and with different structural properties. This is not really acknowledged, though it is appreciated that it would not be possible to deal with all of these different aspects in a first order analysis of this sort.

We appreciate the Reviewer's concern here and have also added an additional caveat about the heterogeneity of urban landscapes and the inability of our large scale approach to resolve this level of detail in Lines 324-327. We have added some more specific details of the UHI data we have compared against as per the source references in Lines 557.

For (b), the authors have not elaborated on important intra-annual contexts for UHI, how the form of precipitation can influence runoff (and related pluvial flooding events).

We are grateful to the Reviewer for the challenge to delve deeper into our results to derive an extra layer of interpretation for intra-annual as opposed to just the long term mean situation for both metrics, and the knock-on implications for flooding and heat stress events. We have now done this additional analysis with the following results which we think greatly strengthens the paper:

“We have focussed so far on mean retention since this is a commonly reported metric in the urban greening literature. However, the temporal sensitivity in performance of urban greening to the interactions of time-variable precipitation and PET forcings may also be important. We have therefore also calculated the seasonality in retention characteristics for all GSOD city locations for the intensive substrate, preferring the point precipitation data over the gridded data owing to the increased importance of accurately reflecting the precipitation intensity variations for this purpose. We find a strong linear relationship between retention seasonality (range of monthly means) and mean retention which holds globally across climate zones (Figure 4). This indicates that regions which are prone to poorer retention performance on average also suffer from a greater variability in performance throughout the year. In contrast, locations with higher mean retention also have more consistent retention performance. This result is consistent with the results noted above where locations with more variable rainfall have generally lower mean retention, with the added nuance that the greater rainfall variability is also reflected in greater seasonal retention variability. This illustrates the importance of careful design of local flood risk management strategies in areas prone to extreme rainfall events beyond the level of detail our large scale results here can inform.

When the impact of seasonality in calculated AET is considered in regard to the cooling potential, the opposite relationship is apparent than for the equivalent retention trend. i.e. we find that locations with high seasonal variability in cooling potential also tend to be those with higher mean cooling potential. This result stems from the interaction of timing differences between any seasonal rainfall and seasonal temperature patterns in a given location i.e. where rainfall and temperature are strongly out of phase with each other, there is generally a smaller variability in cooling potential than where these two climate drivers are in phase with each other. Hence, these results suggest that urban

greening may be more effective in locations suffering from urban heat stress where the wet season also coincides with the summer high temperatures, or where rainfall is evenly distributed throughout the year.

Overall then, the spatial patterns of trade-offs we discover between retention and cooling potential, also translate into the temporal domain. For example, in general terms, high retention-low cooling potential areas will tend to also have more stable performance by both metrics. Conversely, at the other end of the retention-cooling trade-off, more variable performance will generally occur in the low retention-high cooling potential areas (Figure 4).

Figure 4. Relationships between seasonality in retention and cooling potential and the mean value, across the climatic aridity gradient, based on results from the GSOD city locations.

For (c) there is little or no consideration of the ‘green elements’ themselves. For instance, towards the end of the article it becomes obvious that the analysis and framing is based on the very specific context of green roofs (reference to ‘substrate depth’, crop height assumptions (0.12m), and the assumptions of functions being only related to evapotranspiration). This is also clear in the literature used. The extent to which the findings are therefore transferable to ‘urban greening’ in a general sense must therefore be much more carefully considered. For instance, the functional role of urban tree canopy would be expected to differ from the assumptions made here, including the important role of reducing the amount of solar radiation reaching the ground surface (see for instance <https://doi.org/10.1016/j.buildenv.2019.106606>). In my view it is not appropriate to infer to all cases of urban greening interventions. Although the authors make some qualifications in places, e.g. references to ‘ground level parks’ (line 401) in the discussion and ‘engineered green surfaces’ (line 91), these terms are ambiguous. The conclusions need to be tempered accordingly and in my view the paper reframed in the context of the interventions actually being considered. This would help with accessibility, since the terminology used seems strongly related to that used within the green roof literature.

We appreciate the Reviewer’s concern here. We have now made it clearer in the introduction and methods which types of interventions the paper is applicable to and also tempered the conclusions with appropriate caveats as suggested. While our calculation of PET is indeed for well-watered green ‘reference’ vegetation, the model then uses a crop and stress co-efficient approach to model a variety

of plant responses to the prevailing weather and climate in order to estimate AET and drainage. As such, our model results apply to any homogeneously vegetated surfaces from deeper soil profiles common in parks and recreation areas to much more shallow and highly engineered substrates such as green roofs.

As suggested above, at present the claims made in the manuscript are not fully supported by the analysis and require more qualification and explanation to be fully persuasive. This includes making inferences about flooding.

This is a summary of the Reviewers points so far - see our responses above.

Other aspects of the manuscript are also overstated such as the claim in the abstract that the “geographical variation of the relative hydrological and thermal performance benefits of such interventions are unknown”. There are in fact multiple examples of such studies carried out in different climate zones including in the US, Brazil and Europe, some illustrations include DOI: 10.1175/JHM-D-15-0112.1, 10.1016/j.apenergy.2018.03.190. Still, the claim is more suitably qualified in the manuscript itself.

Apologies if our text was not clear enough on this point – we are aware of research on cooling as well as retention of urban greening but our point here is that no studies have addressed their combined performance and the trade-offs between them in general terms. We have changed the text to make this clearer in Lines 61-66 to:

"Despite the overall similarities in the driving processes for potential retention and cooling, no large-scale studies have systematically addressed the assessment of these environmental benefits in combination or their relative performance across different climatic regions. This is of critical importance since, owing to the various potential interactions between different urban greening interventions, to consider one ecosystem service in isolation risks unknowingly providing a potential disservice from another."

Further comparison of results with other studies would be really valuable. Is it possible to carry out some verification of results using examples in the literature? This would make a very good addition to the paper. Scale may well be an issue here, but that also raises questions about the scale of analysis compared to the scale of interventions. The authors touch on some site-level data for Canada.

Again, we are grateful to the Reviewer for the challenge to carry out more evaluation. We have now conducted a detailed second review of the literature specifically to find studies for which sufficient information is available to parametrise our model directly based on information stated in the paper (i.e. with no calibration), as well as retention measurements for multiple events from urban greening interventions. Unsurprisingly, such experimental discharge observations are only widely available specifically for green roof interventions, and we found 13 papers representing a total of 26 different roofs across a diverse range of climates. At each of these locations, we ran the model using gridded ERA5 input data for the specific date range of the individual studies and then compared the mean retention from each study with our model results. The results indicate a very strong correlation between the observed and modelled retention (see below) within the sorts of scatter of observational data expected in such studies. This is a significant additional demonstration that our method can provide robust estimates, even at relatively small scales, of the hydrological functioning of a wide variety of green surfaces in urban areas, despite the coarseness of the forcing data and the inherent assumptions made in the modelling. The methods and results sections have been updated to reflect these additional results.

New addition to Figure S2.

Finally, it would be very interesting to if the authors could elaborate further on the conceptual framework and ‘targeted quantitative guide’ mentioned at lines 31-32. These would be great additions, but it wasn’t quite clear where either were delivered in the current version.

We have now added more discussion of the Budyko framework within which we interpret the results of the study and which makes sense of the broad climatic relationships and, in particular, the importance of aridity, in controlling the effectiveness of urban greening. In terms of the targeted quantitative guide, in addition to making the rasters public available with the paper, we will provide a spreadsheet of results for every global urban area (c. 30 000) which includes our retention and cooling metric results for the different soil profile types with or without irrigation. This will enable easy look-up and comparison between different locations or between different intervention designs within the same location, as a first-order guide to the likely thermal or hydrological efficacy trade-offs. For more research oriented users of the work, we will also provide the Python code for modelling the responses based on local observational data and parameters where these may be available to improve on our coarse-scale estimates.

REVIEWER COMMENTS

Reviewer #1 (Remarks to the Author):

Dear Authors, Your revised manuscript is well-received, and the sense is that we are converging toward resolution of major concerns. Yet, the manuscript requires more work.

Please refer to the marked-up manuscript for detailed comments and edits.

In the view of the reviewer, what is essential here is that the author team be much more careful about what green infrastructure can and cannot do in terms of moderating energy-moisture tradeoffs; the review recommends dropping all mention of groundwater subsidies as irrigation inputs (see. Science 23 April 2021, ppg. 344-345, The hidden crisis below our feet); and better integration of the Budyko hypothesis into the discussion and conclusions.

The author team has provided a valuable, gross geographic scale analysis of tradeoffs that frame the potential for different interventions that may moderate the severity of one energy-moisture scenario or the other. Tree canopy, and green space underlain by thick soil profiles are more likely to be implemented or conserved. The reviewer just does not see engineered green infrastructure as affordable, nor properly implemented at the spatial scales required to affect UHI, much less play a role in moderation of gross-scale energy-moisture balance.

The mention of the urban stream syndrome at the beginning and end of the article suggests its deletion - this is a cross-scale reference that does not make a lot of sense. That is the view of the reviewer, and perhaps the reviewer is missing something here.

Reviewer #3 (Remarks to the Author):

The authors have made some useful modifications to the manuscript and provided further insights in terms of intra-annual variations. This adds some more novelty and interest.

Many of my suggestions have been addressed appropriately and I thank the authors for their considered approach and additional modelling efforts. The expansion of the evaluation section also reassures about the veracity of model outputs.

A few remaining minor concerns are:

I had queried (as had another reviewer) the statement in the abstract "the geographical variation of the relative hydrological and thermal performance benefits of such interventions are unknown" The authors have stressed "we are aware of research on cooling as well as retention of urban greening but our point here is that no studies have addressed their combined performance and the trade-offs between them in general terms" and have made clarifications at lines 61-66. This is fine but the sentence in the abstract needs a similar clarification (lines 17-18).

The caveats of the method have been further elaborated and the authors have stated that they have "now added additional clarification as to the types of green infrastructure we are focussing on in this study in several places in the revised paper". At lines 324-326 the statement has been added "While the model results are appropriate to conditions common in urban parks, recreation areas or engineered green roofs, they are not appropriate to infer to all cases of urban greening interventions." It would be helpful to follow this with the main exclusions before the statement on urban complexities and heterogeneous vegetation density.

While the methods state that no assumption of coverage is made, (comments A96 and A97) and the authors are using dimensionless metrics, it would be helpful to state any underlying assumed unit area for the results implied by the model inputs and related assumptions. For instance this can help to understand the applicability of the results in real world contexts of interventions, and to interpret statements like "Our model results apply to any homogeneously vegetated surfaces from deeper soil profiles common in parks and recreation areas to much more shallow and highly engineered substrates such as green roofs." (lines 72-74). Is there a minimum implied extent threshold for homogeneity of surface properties? This is important given the aim to provide a quantitative guide for urban development, renewal and policymaking, even given the added caveats about additional local scale work and issues.

Possible typographic issues to resolve:

"SMBMs may already be confidently applied in the context of green ground-based urban areas such as parks and recreation areas." (lines 379-380). The meaning of "green ground-based urban areas" wasn't immediately obvious.

"While we recognize that urban areas are incredibly heterogeneous, with vegetation occurring in multiple assemblages, different configurations (e.g. on and under hard landscaping) and with different structural properties. However, our intention here is to provide a first order analysis at large scales and therefore this level of detail cannot be resolved in our approach here." Line 392-396. Delete "While" in sentence 1?

RESPONSE TO REVIEWERS' COMMENTS for:

**Global climate-driven trade-offs between the water retention
and cooling benefits of urban greening**

by: M. O. Cuthbert, G. C. Rau, M. Ekström, D. M O'Carroll, A. J. Bates.

Author responses in blue, line numbers refer to clean revised Manuscript.

Reviewer #1 (Remarks to the Author):

Dear Authors, Your revised manuscript is well-received, and the sense is that we are converging toward resolution of major concerns. Yet, the manuscript requires more work. Please refer to the marked-up manuscript for detailed comments and edits.

Many thanks to the Reviewer for their positive assessment and additional comments. We have made detailed responses in the marked up pdf but also give a summary in line with the Reviewer's summary of main points below.

In the view of the reviewer, what is essential here is that the author team be much more careful about what green infrastructure can and cannot do in terms of moderating energy-moisture tradeoffs;

See response to the Reviewer's penultimate comment below.

the review recommends dropping all mention of groundwater subsidies as irrigation inputs (see. Science 23 April 2021, ppg. 344-345, The hidden crisis below our feet);

We share the Reviewer's concern here and are certainly not recommending groundwater as an irrigation input unless it can be resourced sustainably. However, we are saying that in more arid parts of the world where irrigation would be useful for urban cooling, the most reliable source water supply is often groundwater - this is surely not in doubt in general terms? The key point we then make is that, despite this, groundwater may not be a *sustainable* source of water for this purpose and that this may be a fundamental constraint on the viability of urban cooling. Since irrigation is already commonplace in dryland cities, rather than avoiding this issue, we feel this issue needs addressing head-on. Hence we have re-written the paragraph to try and make the cautionary message stronger and also added mention of sustainability in the abstract. We hope this is in the spirit of the Reviewer's concern.

and better integration of the Budyko hypothesis into the discussion and conclusions.

We agree and have now added a statement on the Budyko hypothesis in the discussion section.

The author team has provided a valuable, gross geographic scale analysis of tradeoffs that frame the potential for different interventions that may moderate the severity of one energy-moisture scenario or the other. Tree canopy, and green space underlain by thick soil profiles are more likely to be implemented or conserved. The reviewer just does not see engineered green infrastructure as affordable, nor properly implemented at the spatial scales required to affect UHI, much less play a role in moderation of gross-scale energy-moisture balance.

While we agree that it is important to consider the cost-effectiveness of engineered green infrastructure at scale, it is beyond the scope of our analysis to include a quantitative analysis on this matter but have now made some comments to this effect in the discussion.

The mention of the urban stream syndrome at the beginning and end of the article suggests its deletion - this is a cross-scale reference that does not make a lot of sense. That is the view of the reviewer, and perhaps the reviewer is missing something here.

Since the paper is about the trade-offs between two key aspects of urban greening, we feel it important to also mention the importance of other trade-offs such as those mentioned regarding the USS which may need to be considered when analysing the impact of urban greening design in a more holistic sense. No changes made.

Reviewer #3 (Remarks to the Author):

The authors have made some useful modifications to the manuscript and provided further insights in terms of intra-annual variations. This adds some more novelty and interest. Many of my suggestions have been addressed appropriately and I thank the authors for their considered approach and additional modelling efforts. The expansion of the evaluation section also reassures about the veracity of model outputs.

Many thanks to the Reviewer for their positive assessment and additional comments.

A few remaining minor concerns are:

I had queried (as had another reviewer) the statement in the abstract “the geographical variation of the relative hydrological and thermal performance benefits of such interventions are unknown” The authors have stressed “we are aware of research on cooling as well as retention of urban greening but our point here is that no studies have addressed their combined performance and the trade-offs between them in general terms” and have made clarifications at lines 61-66. This is fine but the sentence in the abstract needs a similar clarification (lines 17-18).

We agree and have now clarified this in the abstract by adding the word ‘combined’.

The caveats of the method have been further elaborated and the authors have stated that they have “now added additional clarification as to the types of green infrastructure we are focussing on in this study in several places in the revised paper”. At lines 324-326 the statement has been added “While the model results are appropriate to conditions common in urban parks, recreation areas or engineered green roofs, they are not appropriate to infer to all cases of urban greening interventions.” It would be helpful to follow this with the main exclusions before the statement on urban complexities and heterogeneous vegetation density.

We agree and have now added some examples of exclusions at the end of the relevant sentence.

While the methods state that no assumption of coverage is made, (comments A96 and A97) and the authors are using dimensionless metrics, it would be helpful to state any underlying assumed unit area for the results implied by the model inputs and related assumptions. For instance this can help to understand the applicability of the results in real world contexts of interventions, and to interpret statements like “Our model results apply to any homogeneously vegetated surfaces from deeper soil

profiles common in parks and recreation areas to much more shallow and highly engineered substrates such as green roofs.” (lines 72-74). Is there a minimum implied extent threshold for homogeneity of surface properties? This is important given the aim to provide a quantitative guide for urban development, renewal and policymaking, even given the added caveats about additional local scale work and issues.

While our model assumes homogeneity in vegetation type it could, in principle at least, be applied in places where the heterogeneity of the land-cover is adequately known to enable an equivalent parameter approach to be applicable. However, given the coarseness of the present study, we feel it would be misleading here to attempt to put a precise constraint on the exact limit of the spatial scale of land-cover heterogeneity that could be accommodated using this approach.

Possible typographic issues to resolve:

“SMBMs may already be confidently applied in the context of green ground-based urban areas such as parks and recreation areas.” (lines 379-380). The meaning of “green ground-based urban areas” wasn’t immediately obvious.

We agree and have now clarified this to say ‘deeply-rooted vegetated urban areas such as parks and recreation areas’.

“While we recognize that urban areas are incredibly heterogeneous, with vegetation occurring in multiple assemblages, different configurations (e.g. on and under hard landscaping) and with different structural properties. However, our intention here is to provide a first order analysis at large scales and therefore this level of detail cannot be resolved in our approach here.” Line 392-396. Delete “While” in sentence 1?

We agree and have now deleted the word ‘while’.

REVIEWER COMMENTS

Reviewer #1 (Remarks to the Author):

Dear Authors, Thanks so much for your revision. This manuscript reads very well, and I find that my concerns have been fully addressed. I recommend acceptance of this manuscript. Best of luck in the rest of the publication process, and thank you for your important contribution.

Reviewer #3 (Remarks to the Author):

Most comments have been addressed adequately, thank you.

However, the response to my query about scale assumptions is very unclear. If the study is too coarse to clarify the assumptions behind the statement "Our model results apply to any homogeneously vegetated surfaces" then it is difficult to envisage how other claims are supported, specifically the applicability of findings to specific cases of green roofs and parks. If that isn't clarified then readers will struggle to interpret what the results mean in real world contexts, as one of the stated goals of the paper as a whole.

RESPONSE TO REVIEWERS' COMMENTS for:

Global climate-driven trade-offs between the water retention and cooling benefits of urban greening

by: M. O. Cuthbert, G. C. Rau, M. Ekström, D. M O'Carroll, A. J. Bates.

Author responses in blue, line numbers refer to clean revised Manuscript.

Reviewer #1 (Remarks to the Author):

Dear Authors, Thanks so much for your revision. This manuscript reads very well, and I find that my concerns have been fully addressed. I recommend acceptance of this manuscript. Best of luck in the rest of the publication process, and thank you for your important contribution.

Excellent, many thanks to the Reviewer for their detailed comments which have certainly strengthened the paper.

Reviewer #3 (Remarks to the Author):

Most comments have been addressed adequately, thank you.

Great. Again, many thanks to the Reviewer for their detailed comments which have certainly strengthened the paper.

However, the response to my query about scale assumptions is very unclear. If the study is too coarse to clarify the assumptions behind the statement "Our model results apply to any homogeneously vegetated surfaces" then it is difficult to envisage how other claims are supported, specifically the applicability of findings to specific cases of green roofs and parks. If that isn't clarified then readers will struggle to interpret what the results mean in real world contexts, as one of the stated goals of the paper as a whole.

Apologies - on re-reading their original review comment and our response, the additional request for clarification is welcome. Essentially, in terms of the scale-applicability of our model to real world urban greening scenarios, as long as the application has a substrate similar to the ranges assumed by the model, and that the local climate is not significantly different to that of the climate simulated by ERA5, then the results in the paper are representative down to a point scale. This is already communicated to some extent in the methods where we show the success of the model to replicate results of a range of small scale green roof studies:

L489: This is a significant additional demonstration that our method can provide robust estimates, even at relatively small scales, of the hydrological functioning of a wide variety of green surfaces in urban areas, despite the coarseness of the forcing data and the inherent assumptions made in the modelling.

However, to strengthen this point we have now also added the following statement (blue text) in the final paragraph of the main text to try and resolve any remaining concern:

“While the model results are appropriate to conditions common in urban parks, recreation areas or engineered green roofs, they are not appropriate to infer to all cases of urban greening interventions (e.g. living walls or pillars, dispersed shrub or tree planting). The gross landform and land cover of a given cityscape may add complexities not captured in our coarse-scale global analysis, e.g. due to variations in city morphology and anthropogenic heat contributions, or more complex short term (e.g. diurnal) temporal dynamics or spatial effects of heterogeneous vegetation density³⁴. However, our results are directly applicable to real world urban greening scenarios even down to small, ‘point’, scale, as long as the application substrate is sufficiently analogous to one of those assumed by our model, and that the local climate is not significantly different to that of the climate simulated by ERA5. Since the necessary high-resolution data and modelling are not readily available for most cities around the world, our framework can therefore provide a first-order guideline to inform generalized or large-scale strategies for urban development and renewal. We have provided our results as a lookup table (link to download) for all global city areas, as a resource for such first-pass assessment as part of a tiered approach to urban development strategy until more local data or models are available with which to assess the concept at local-scales for a particular city. Furthermore, our data can be used to explore how the performance of a green infrastructure in a current climate may perform under future climate conditions by finding a spatial analogue for a projected climate future (i.e a space-for-time substitution).”

We hope that clarifies the Reviewer’s remaining concern.

REVIEWERS' COMMENTS

Reviewer #3 (Remarks to the Author):

The further clarification points readers to the model assumptions and helps them to understand that the model cannot be applied in real world fine scale situations unless specific conditions are met. Thanks for giving this further consideration.